# Industrial Calibration Procedure for Confocal Microscopes

**DOI:** 10.3390/ma12244137

**Published:** 2019-12-10

**Authors:** Alberto Mínguez Martínez, Jesús de Vicente y Oliva

**Affiliations:** 1Laboratorio de Metrología y Metrotecnia (LMM), Escuela Técnica Superior de Ingenieros Industriales (ETSII), Universidad Politécnica de Madrid (UPM), c./José Gutiérrez Abascal, 2, 28006 Madrid, Spain; 2Centro Láser, Universidad Politécnica de Madrid (UPM), Campus Sur, Edificio “La Arboleda”, c./Alan Turing, 1, 28031 Madrid, Spain

**Keywords:** coordinate metrology, confocal microscopy, measurement, calibration, traceability, uncertainty, quality assessment

## Abstract

Coordinate metrology techniques are widely used in industry to carry out dimensional measurements. For applications involving measurements in the submillimeter range, the use of optical, non-contact instruments with suitable traceability is usually advisable. One of the most used instruments to perform measurements of this type is the confocal microscope. In this paper, the authors present a complete calibration procedure for confocal microscopes designed to be implemented preferably in workshops or industrial environments rather than in research and development departments. Therefore, it has been designed to be as simple as possible. The procedure was designed without forgetting any of the key aspects that need to be taken into account and is based on classical reference material standards. These standards can be easily found in industrial dimensional laboratories and easily calibrated in accredited calibration laboratories. The procedure described in this paper can be easily adapted to calibrate other optical instruments (e.g., focus variation microscopes) that perform 3D dimensional measurements in the submillimeter range.

## 1. Introduction

In industry, coordinate measuring machines (CMMs) are widely used to carry out dimensional measurements, because these kinds of machines are capable of measuring many different types of geometries with great flexibility and sufficient accuracy. For this reason, CMMs can be considered as universal measuring devices [1]. In addition, modern advanced manufacturing processes demand a deep study of surface textures. Probably the most common method for surface texture verification up to recent years was to perform roughness measurements with a roughness measuring machine—usually a 2D stylus instrument [2]. In many cases, manufacturers prefer to verify texture and geometry without mechanical contact between the instrument and surface [3]. Due to this, optical instruments for coordinate metrology have been developed. ISO 25178-6 lists optical methods for measuring surface texture [4], and among them, it includes confocal microscopy, which permits both dimensional and 2D/3D roughness measurements [5] without mechanical contact. 

The confocal microscope was developed in 1955 by Minsky [6,7] and allows images of optical sections of samples to be obtained, from which the full 3D object geometry can be reconstructed. Confocal microscopy is also important, because it is a powerful tool for observation and measurement at both scientific research and workshop levels. It presents the following advantages [8]:It adds the Z-axis to traditional measuring optical microscopes, which only work in the XY plane.It allows analysis of the 3D geometry of the object surface and characterization of its quality from data points acquired while scanning it.Its lateral resolution is better than in traditional optical microscopy.It permits more precise 3D images of the objects being measured to be obtained that are of higher quality and in less time compared to other methods. This allows many useful measurements to be carried out in short intervals of time.Transparent specimens can be observed, as can sections with a certain thickness, without the need to section the object under study.

Confocal microscopy has applications in many fields, both in research and industrial applications. This type of microscope is widely used in biomedical science, material science, and surface quality metrology at micro and macro scales [9]. However, there is no standardized procedure to calibrate and provide traceability to these instruments in the fields of coordinate metrology and surface texture metrology. Intense work is being done around ISO standard 25178-700, but it is still in the draft stage. The calibration of measurement instruments is crucial to maintain the traceability of measurement results [10]. As is widely known, traceability can be defined as the property of a measurement result by which it can be related to a reference through an uninterrupted and documented chain of calibrations, each of which contributes to measurement uncertainty [11]. 

The purpose of this paper is to describe a way to provide suitable traceability to a confocal microscope when performing metrological activities using single topography measurements, in the fields of both coordinate metrology and roughness metrology. Please note that when image stitching is not used (single topography), there is no movement of the XY stage, and there is therefore no need to calibrate the displacements of this stage. The calibration procedure presented by the authors is intended to be simple and is based on classical mechanical standards. Note that the objective was not to perform a state-of-the-art calibration of a confocal microscope [12,13,14,15], neither was it to achieve very low uncertainties; the objective was to ensure adequate traceability with adequate uncertainty estimation in the field of dimensional metrology in the submillimeter range.

Prior to carrying out the calibration, an analysis of the operating principle of the confocal microscope is necessary for a better understanding of the device. This kind of microscope usually uses a low-power, high-intensity, monochromatic laser system for illumination [16,17,18,19]. A laser beam passes through a beam splitter, and one of the beams is then redirected to the sample, passing through complex optics [7]. Once the scanning surface is illuminated, the reflected beam travels back along the same path. If the illumination is properly focused on the surface, the reflected beam will go to the detector without losing intensity, but if the surface is out of focus, the intensity will be lower. The filtered beam arrives at the detector and a computer system processes the signal, making a 3D reconstruction of the surface [9,16,18].

Several factors affect the quality of these measurements [5,20]: Metrological characteristics of the instrument: measurement noise, flatness deviation, non-linearity errors, amplification coefficients, and perpendicularity errors between axes.Instrument geometry: alignment of components and the XY stage and rotary stage error motions.Source characteristics: focal spot size and drift.Detector characteristics: pixel response, uniformity and linearity, detector offset, and bad pixels.Reconstruction and data processing: surface determinations, data representation, and calculation approaches.Environmental conditions: temperature, humidity, and vibration.

As can be seen in Figure 1, the confocal microscope projects, through a complex optical system, illumination patterns over the surface that is being explored and captures the returned beam through the same pattern of illumination. As a result, it is possible to discriminate if the returned beams are out of focus and filter them [5,7,16,17,21]. In Figure 2, this property is shown in detail.

Once the in-focus image goes to the detector, the computational treatment starts. The electronic controller moves the objective along the Z-axis in order to permit the confocal microscope to capture 2D images at different 
z
 coordinates. As 2D images are composed of pixels, 3D images obtained with confocal microscopes are composed of voxels, as shown in Figure 3. If an interpolation if made between consecutive 2D images, it is possible to create a 3D model of the scanned surface [23].

In order to achieve dimensional traceability in dimensional measurements carried out with confocal microscopes, it is necessary to perform a 3D calibration of the instrument. This calibration should provide estimations of voxel sizes along X, Y, and Z axes, but it would also be advisable to provide estimations of perpendicularity errors between axes. Additionally, as the confocal microscope can be used to perform roughness measurements, a specific calibration of the instrument for roughness measurements is advisable.

## 2. Materials and Methods

In order to ease the understanding of the calibration procedures described later on, a calibration example was carried with the following confocal microscope and software:Leica DCM3D confocal microscope (Wetzlar, Germany) with a 10× objective (EPI-L, NA = 0,30). Field of view 1270 µm × 952 µm (768 × 576 pixels); 1.65 µm nominal voxel width. The overall range of the Z-axis is 944 µm using 2 µm axial steps (voxel height), but the instrument is used in a reduced working range of only 100 µm.SensoSCAN—LeicaSCAN DCM3D 3.41.0 software developed by Sensofar Tech Ltd. (Terrassa, Spain).

In this paper, we propose a calibration procedure that is only valid for single topography measurements—that is, without using image stitching. In single topography measurements, the XY stage is not moved during measurement, and its errors do not contribute to uncertainty. When using image stitching (extended topography measurements), the XY stage does contribute to uncertainty, and the calibration procedure described in this paper should be updated using techniques such as those described in [14,15]. The complete calibration procedure includes the following:Calibration of the X and Y scales, using a stage micrometer as a reference measurement standard.Estimation of perpendicularity error between X and Y axes.Estimation of the flatness deviation of the focal plane using an optical flat.Calibration of Z scale using a calibrated steel sphere.Calibration of the confocal microscope for the measurement of 2D roughness using periodic and aperiodic 2D roughness measurement standards.All uncertainties are estimated following the mainstream GUM method (Guide to the Expression of Uncertainty in Measurement [24]) or EA-04/02 M:2013 document [25], as they are standard procedures in calibration laboratories accredited under ISO 17025 [26].

All reference measurement standards used were chosen to be:Easy to find.Easy to calibrate with low enough uncertainties in National Measurement Institutes (NMIs) or preferably in accredited calibration laboratories (ACLs).Stable mechanical artifacts that could guarantee long recalibration intervals.Common in the field of dimensional metrology in order to facilitate their acquisition, calibration, and correct use.

### 2.1. Flatness Verification

Before calibrating the X and Y axes, a flatness verification must be performed. This is necessary because it is often observed that the XY plane of confocal microscopes is slightly curved. This is evident when exploring a flat surface such as an optical flat whose total flatness error is usually lower than 50 nm. In these cases, the reference flat surface when observed by the confocal microscope appears curved, as if it was a cap of a sphere or an ellipsoid. According to manufacturers, this error is usually small enough, but it is impossible to carry out an accurate measurement without taking this component of uncertainty into account [27]. 

For this verification, the authors propose following a procedure based on [28], but using a confocal microscope instead of an interferometer. The software of the confocal microscope provides a topographic map of the explored surface from which the total flatness error (peak to peak) or the RMS (root mean square error) flatness can be estimated.

The calibration is done in two positions (0° and 90°) and therefore, two measurements are obtained (Figure 4):

For this calibration, the authors recommend using the RMS flatness deviation, because it is more statistically stable than the total flatness deviation. As discussed in [29] (Section 1.3.5.6 “Measures of Scale”), the total flatness deviation—which is equivalent to the range—is very sensible to the presence of outliers because it is determined as the difference between the two most extreme points. The problem increases with the number 
N
 of points from which the range is determined. In our case, the number of points was very large (
N=768×576=442,368)
. For these reasons, other parameters such as the standard deviation (equivalent to the root mean square error), or even better, the median absolute deviation (MAD) should be used. Since confocal software always includes the RMS flatness deviation and it is not very easy to include MAD, we recommend the RMS flatness deviation. 

### 2.2. XY Plane Calibration

In the literature, it is possible to find several procedures for this calibration. Following the studies of de Vicente et al. [30] and Guarneros et al. [31], it is possible to calibrate X and Y scales and estimate their perpendicularity error by making measurements of a stage micrometer in four positions (Figure 5).

A stage micrometer is easy to calibrate in a National Measurement Institute (NMI) or in an accredited calibration laboratory (ACL) with sufficiently small uncertainty (equal to or lower than 1 µm) for the calibration of a confocal microscope. 

It is strongly recommended that the stage micrometer should be metallic and have the marks engraved, not painted, such as those used to calibrate metallographic microscopes. Marks painted over glass are difficult to detect with a confocal instrument.

The matrix model proposed for calibration by de Vicente et al. [32] is as follows:
(1)
$$\left[\begin{array}{c} x\\ y \end{array}\right]=\left[\begin{array}{c} p\\ q \end{array}\right]+\left[\begin{array}{cc} c_{xy}+a & \theta /2\\ \theta /2 & c_{xy}-a \end{array}\right]\cdot \left[\begin{array}{c} p\\ q \end{array}\right],$$

where 
(p,q) 
are the readings directly provided by the confocal microscope for the Cartesian coordinates in the XY plane. 
(x,y)
 are the corrected Cartesian coordinates once the calibration parameters 
$c_{xy}$
, 
a,
 and 
θ
 have been applied using the previous matrix model.

The meanings of these three parameters are as follows:


$c_{xy}$
 represents the deviation of actual pixel width 
$w_{xy}$
 from the nominal pixel width 
$w_{xy,nom}$
:
(2)
$$w_{xy}=w_{xy,nom}\cdot \left(1+c_{xy}\right).$$



a
 represents the difference between pixel widths along X-axis (
$w_{x}$
) and Y-axis (
$w_{y}$
):
(3)
$$w_{x}=w_{xy,nom}\cdot \left(1+c_{xy}+a\right),$$



(4)
$$w_{y}=w_{xy,nom}\cdot \left(1+c_{xy}-a\right),$$



(5)
$$w_{xy}=\frac{\left(w_{x}+w_{x}\right)}{2}.$$



θ
 represents the perpendicularity error between the X-axis and Y-axis. The actual angle between these axes is 
π/2−θ
.

The amplification coefficients
$\;\alpha_{x}$
, 
$\alpha_{y}$
, and 
$\alpha_{z}$
 of the axes (according to ISO 25178-70 [33]) are:
(6)
$$\alpha_{x}=1+c_{xy}+a,$$



(7)
$$\alpha_{y}=1+c_{xy}-a,$$



(8)
$$\alpha_{z}=1+c_{z}.$$


We recommend using the average pitch 
ℓ
 of the stage micrometers. 
ℓ
 is the average of all individual pitches (distances between two consecutive marks) observed in the images provided by the confocal microscope. Figure 6 summarizes the measurement of the stage micrometer in one position. Using special software for this task, written in Matlab® R2019a and developed at the Laboratorio de Metrología y Metrotecnia (LMM), it is possible to automatically detect and estimate the distance 
$d_{i}$
 of each mark from the zero mark. Using this software, all the distances (pitches) between two consecutive marks in the stage micrometers were estimated (Figure 6a). Moreover, pitches can be measured in different positions—in the middle and in higher and lower positions—which permits the estimation of the repeatability during the pitch measurements. In Figure 6a, for each pitch, the average value is represented by a circle, and the measurement variability around this value is represented with a vertical line. In order to estimate the non-linearity errors 
$e_{i}$
 (Figure 6b), a straight line 
$d_{i}≅m+\ell \cdot i$
 was fitted, and the errors were estimated as 
$e_{i}=d_{i}-\left(m+\ell \cdot i\right)$
. The coefficient 
m
 represents the deviation of the zero mark from its estimated position and 
ℓ
 represents the average pitch.

If the previously mentioned special software is not available, measurements of distances between marks must be done by hand. However, although it is more laborious, the entire procedure described above for the estimation of the average pitch 
ℓ
 and non-linearity errors 
$e_{i}$
 can be carried out without any problem.

Let 
$\ell_{0}$
 be the average pitch of the stage micrometer certified by a suitable laboratory with a standard uncertainty 
$u(\ell_{0})$
. 
$\ell_{1}$
, 
$\ell_{2}$
, 
$\ell_{3}$
, and 
$\ell_{4}$
 are the average pitches measured with the confocal microscope in positions 0°, 90°, 45°, and 135°, respectively. Their corresponding standard uncertainties are 
$u\left(\ell_{1}\right)$
, 
$u\left(\ell_{2}\right)$
, 
$u\left(\ell_{3}\right)$
, and 
$u\left(\ell_{4}\right),$
 where only the variability observed in Figure 6 (or equivalent ones) was taken into account. 

When the matrix model is applied to positions 0°, 90°, 45°, and 135°, we obtain the following expressions that permit simple estimations of calibration parameters 
$c_{xy}$
, 
a,
 and 
θ
:
(9)
$$\mathrm{Position}\;0{}^{\circ}:\;\ell_{1}\cdot \left(1+c_{xy}+a\right)≅\ell_{0},$$



(10)
$$\mathrm{Position}\;90{}^{\circ}:\;\ell_{2}\cdot \left(1+c_{xy}-a\right)≅\ell_{0},$$



(11)
$$\mathrm{Position}\;45{}^{\circ}:\;\ell_{3}\cdot \left(1+c_{xy}+\frac{\theta }{2}\right)≅\ell_{0},$$



(12)
$$\mathrm{Position}\;135{}^{\circ}:\;\ell_{4}\cdot \left(1+c_{xy}-\frac{\theta }{2}\right)≅\ell_{0}.$$


From these expressions, it is easy to conclude that possible estimations of
$\;c_{xy}$
,
 a, 
and
 θ 
are:
(13)
$$c_{xy}=\frac{\ell_{0}}{4}\cdot \left(\frac{1}{\ell_{1}}+\frac{1}{\ell_{2}}+\frac{1}{\ell_{3}}+\frac{1}{\ell_{4}}\right)-1,$$



(14)
$$\mathrm{with}\;u\left(c_{xy}\right)=\frac{\sqrt{u^{2}\left(\ell_{0}\right)+\left[u^{2}\left(\ell_{1}\right)+u^{2}\left(\ell_{2}\right)+u^{2}\left(\ell_{3}\right)+u^{2}\left(\ell_{4}\right)\right]/16}}{\ell_{0}},$$



(15)
$$a=\frac{\ell_{0}}{2}\cdot \left(\frac{1}{\ell_{1}}-\frac{1}{\ell_{2}}\right),$$



(16)
$$\mathrm{with}\;u\left(a\right)=\frac{\sqrt{u^{2}\left(\ell_{1}\right)+u^{2}\left(\ell_{2}\right)}}{2\ell_{0}}.$$



(17)
$$\theta =\ell_{0}\cdot \left(\frac{1}{\ell_{3}}-\frac{1}{\ell_{4}}\right),$$



(18)
$$\mathrm{with}\;u\left(\theta \right)=\frac{\sqrt{u^{2}\left(\ell_{3}\right)+u^{2}\left(\ell_{4}\right)}}{\ell_{0}}.$$


Correlations between these parameters (
$c_{xy}$
,
 a, 
and
 θ
) are usually very small (lower than 0.01). Therefore, these correlations can be neglected.

### 2.3. Z-Axis Calibration

Document [34] proposes calibrating the Z-axis using a step gauge built with gauge blocks over an optical flat (Figure 7). However, the short field of view of confocal microscopes makes it difficult to carry out the calibration with this type of measurement standard.

To solve this problem, several authors [5,35] have proposed the use of step height standards (Figure 8). Wang et al. [35] used them with the nominal values 24, 7, 2, and 0.7 µm. This kind of measurement standard has several grooves whose nominal depths cover the range of use of the confocal microscope on the Z-axis. Following their procedures, every groove has to be measured 10 times, changing the position of the standard on the objective. 

This kind of standard is typically used for roughness calibration. If the purpose is to make a calibration on the Z-axis, these standards have the limitation of the groove depth, which usually is small to cover the range of the Z-axis.

In order to solve this problem, we propose using a small metallic sphere, as shown in Figure 9, with a nominal diameter between 1 and 10 mm, similar to the one used in [38]. This kind of measurement standard is easy to find and easy to calibrate in both NMIs and ACLs with uncertainties equal to or lower than 0.5 µm. The software of confocal microscopes usually permits a spherical surface to be fit to the points detected over the surface of the spherical measurement standard. If not, coordinates 
(p,q,r)
 obtained with the confocal microscope can be exported to a text file and processed with a routine similar to that described in Appendix A. Therefore, it is possible to compare the certified diameter 
$D_{0}$
 of the sphere against the diameter 
$D_{m}\;$
of the spherical surface fitted by the confocal microscope. We propose the use of an extended matrix model to take into account the calibration of the Z-axis:
(19)
$$\left[\begin{array}{c} x\\ y\\ z \end{array}\right]=\left[\begin{array}{ccc} 1+c_{xy}+a & \theta /2 & 0\\ \theta /2 & 1+c_{xy}-a & 0\\ 0 & 0 & 1+c_{z} \end{array}\right]\cdot \left[\begin{array}{c} p\\ q\\ r \end{array}\right]$$

where 
p,  q,  and r
 are readings provided by the confocal microscope for the Cartesian coordinates
 x,y, and z
. The calibration parameters are those described in Section 2.2 (
$c_{xy},a,\theta$
), and the new parameter 
$c_{z}$
 is introduced to permit the calibration in the Z-axis. The corrected 
z
 coordinate is:
(20)
$$z=\left(1+c_{z}\right)\cdot r.$$


This simple matrix model supposes that there is no (or negligible) perpendicular error between the Z-axis and the XY plane. This hypothesis is very close to reality when the Z-axis range is clearly lower than ranges of the X and Y axes. When the Z-axis range is equal to or greater than X and Y ranges, a more complex model must be used (zero terms in the matrix of the model are no longer zero—see, for example, [39]). It is easy to demonstrate that, using the matrix model, the corrected diameter 
D
 of the spherical surface fitted by the confocal microscope software is:
(21)
$$D=D_{m}\cdot \frac{1+2c_{xy}}{1+c_{z}}$$

where 
$D_{m}\;$
is the diameter provided by the confocal microscope prior to applying any calibration parameter. Therefore, an estimation of 
$c_{z}$
 is

(22)
$$c_{z}=\frac{D_{m}}{D_{0}}\cdot \left(1+2c_{xy}\right)-1$$

where 
$D_{0}$
 is the certified diameter of the sphere by the ACL. The standard uncertainty of 
$c_{z}$
 is


(23)
$$u(c_{z})=\sqrt{\frac{u^{2}(D_{0})+u^{2}(D_{m})}{D_{0}^{2}}+4u^{2}\left(c_{xy}\right)}.$$


Equation (22) of 
$c_{z}$
 shows a clear positive dependency with
$\;c_{xy}$
. Therefore, the correlation coefficient 
$r\left(c_{z},c_{xy}\right)$
 should be estimated, and it can be done using the following expression:
(24)
$$r\left(c_{z},c_{xy}\right)=2\cdot \frac{u\left(c_{xy}\right)}{u\left(c_{z}\right)}.$$


Note that the correlation coefficient is denoted as 
$r\left(c_{z},c_{xy}\right)$
. Do not confuse it with the reading of the confocal microscope for the Z-axis, which is denoted as 
r
.

### 2.4. Calibration for Roughness Measurements

The calibration of the Z-axis against the reference sphere (previous section) guarantees the traceability of the vertical measurements performed with the confocal microscope to the SI unit of length (the meter). Therefore, any vertical roughness parameter will have an adequate traceability once the instrument has been calibrated along its Z-axis. Notwithstanding, we followed the recommendation included in documents DKD-R 4-2 [40,41,42], which propose performing an additional calibration against roughness standards to validate the Z-axis calibration for roughness measurements.

Many parameters are used to characterize surface texture. Among the 2D roughness parameters, one of the most widely used is the 
$R_{a}$
 parameter, which is the arithmetic mean of the absolute values of the profile deviations from the mean line of the roughness profile [43]. We only consider the
$\;R_{a}\;$
parameter during calibration, but readers interested in other 2D roughness vertical parameters (
$R_{q}$
, 
$R_{p}$
, 
$R_{v}$
, 
$R_{z}$
, …) can use the same calibration procedure described in this paper but with minor variations. Calibration was performed in the range 
$0.1<R_{a}\leq 2$
 µm. For this range, according to ISO 4288 [44], the sampling length should be 
$l_{r}$
 = 0.8 mm, which is possible to carry out with a field of view of 1270 µm × 952 µm. For 
$R_{a}>2$
 µm, the sampling length should be 
$l_{r}$
 = 2.5 mm or higher, and it is impossible to achieve this with a field of view of 1270 µm × 952 µm (10× objective). It makes no sense to measure roughness lower than 
$R_{a}=0.1\;$
 µm with an instrument with repeatability in the Z-axis of around 0.5 µm. Therefore, calibration for 
$R_{a}<0.1$
 µm and 
$R_{a}>2$
 µm was discarded.

Figure 10a shows three metallic, aperiodic 2D roughness standards. Figure 10b shows three glass, periodic 2D roughness standards. Other types of 2D roughness profile types are described in Section 7 of ISO 25178-70 [33]. We recommend the use of aperiodic standards because they cover a wide range of wavelengths, in contrast to periodic standards that only cover a single wavelength. However, periodic standards were used in this case in order to complete the range of measurements between 0.1 µm and 2 µm for different calibration points and for different materials (glass instead of metallic items).

These standards were measured over five different zones in two different orientations (Figure 11a,b). In each zone, the measurement was carried out along a line perpendicular to the roughness lines (Figure 11c) and located at the center of the zone. Therefore, a total of 
2×5=10
 roughness measurements were obtained from each standard. 

We recommend using at least three different roughness standards with nominal values of 
$R_{a}$
 uniformly distributed along the range where the instrument must be calibrated. However, it is advisable to use five or more standards and, if possible, standards made of different materials (e.g., metallic and glass).

It is important to note that there will be differences between measurements obtained with a confocal microscope and measurements obtained with a stylus instrument [4,45].

The main reasons for this are as follows.The way the surface is detected is totally different: microscopes use light, and stylus instruments use a mechanical tip. Usually, optical instruments present higher instrument noise than stylus instruments. Possible reasons are the effects of multiple scattering and discontinuities [45]. As a consequence, optical instruments tend to overestimate surface roughness.Stylus instruments permit evaluation lengths 
$l_{n}$
 that are as long as necessary (see ISO 4288 [44]). Microscopes usually have small fields of view that limit the maximum length of the profile that can be scanned. For example, for samples with 0.1 µm 
$<R_{a}\leq$
 2 µm, ISO 4288 recommends using five sampling lengths 
$l_{r}=0.8$
 mm for a total evaluation length 
$l_{n}=4$
 mm. This is not a problem for stylus instruments, which can cope with longer evaluation lengths (up to 100 mm in some cases). However, the confocal microscope described at the beginning of Section 2 has a maximum evaluation length of 1.27 mm. Therefore, only one sampling length 
$l_{r}=0.8$
 mm could be used. Using only one sampling length instead of five usually causes a bias toward lower 
$R_{a}$
 values, which are accompanied by an increase in variability. The effect is considerably higher when even the sampling length 
$l_{r}$
 has to be reduced.

In order to ensure a good match between roughness measurements performed with stylus instruments and optical instruments, the concept of “bandwidth matching” should be correctly applied. This term refers to the good correspondence between the spectral bandwidths of two different instruments used for roughness measurements [45]. The effective spectral bandwidth of a roughness-measuring instrument is limited by the two cut-off wavelengths, 
$\lambda_{S}$
 (a high-pass filter), and 
$\lambda_{C}$
 (a low-pass filter), and it is influenced by the X-axis resolution and tip radius in stylus instruments or lateral resolution and pixel size in optical instruments. 

### 2.5. Summary of Characteristics of Measurement Standards Used during Calibration

In this section, the nominal values and the uncertainties of the different reference measurement standards used during calibration are summarized. All of them were calibrated in ACLs.

We include the calibration of the confocal microscope against roughness standard #6 only for informative purposes. Its measurements were made using a sampling length of 
$l_{r}=0.8$
 mm because of the reduced field of view of the instrument (with an 10× objective). However, ISO 4288 [44] recommends the use of a sampling length 
$l_{r}=2.5$
 mm, which is a measurement that is impossible to achieve with a 10× objective.

## 3. Results

### 3.1. Flatness Verification

The following figure shows a topographic image of the optical flat of Figure 4 that was used as a flatness calibration surface. This optical flat was previously calibrated in an accredited laboratory. The total flatness error was 118 nm with a standard uncertainty of 25 nm (
k=1
), and its RMS flatness was 28 nm with a standard uncertainty of 7 nm (
k=1
); see Table 1.

Figure 12 shows the absence of significant curvature in the XY plane. A slight uncorrected curvature of about 0.6 µm (peak to peak) was observed, which is small and could be neglected when compared with the Z-axis axial step (2.0 µm) and observed instrument noise (about 1.0 µm peak to peak). This could be an empirical demonstration of a good adjustment and/or correction of the microscope by the manufacturer. In a situation such as this, there is no need to apply any further correction to compensate the curvature of the XY plane.

Table 2 shows the results of the measurements performed with the confocal microscope (in both positions 0° and 90°).

The RMS values of Table 2 are small when compared with the Z-axis axial step of 2.0 µm. Therefore, they were probably caused by the lack of repeatability of the instrument. In any case, the most conservative option is to estimate a component of the uncertainty associated with the possible curvature of the XY plane equal to the average value of both RMS values of Table 2:
(25)
$$u_{\mathrm{FLT}}=0.54\;µm$$


A better estimation for 
$u_{\mathrm{FLT}}$
 would likely be to quadratically subtract the RMS flatness of the optical flat (
0.28
 µm):
(26)
$$u_{\mathrm{FLT}}=\sqrt{{\left(0.54\;\mu m\right)}^{2}-{\left(0.028\;\mu m\right)}^{2}}=0.539\;µm$$


Regardless, we considered that the first estimation (
$u_{\mathrm{FLT}}=0.54$
 µm) is slightly more conservative and clearly simpler.

### 3.2. XY Plane Calibration

Figure 13 shows the four positions (0°, 45°, 90°, 135°) in which the stage micrometer was measured in the confocal microscope during the XY plane calibration.

In each position, the average pitch 
$\ell_{i}$
 was determined from readings provided by the confocal microscope. The results are shown in Table 3. 

The average value for repeatability in the XY plane was 
$s_{r}\left(x\right)=s_{r}\left(y\right)$
 = 0.4 µm. This is a reasonable value when compared with the 1.65 µm lateral resolution (nominal voxel width).

The stage micrometer had a certified average pitch 
$\ell_{0}$
 = 9.980 µm with a standard uncertainty 
$u(\ell_{0})$
 = 0.005 µm.

Using the expression for Section 2.2, we obtained the following estimations for calibration parameters
$\;c_{xy}$
, 
a,
 and
 θ
:
(27)
$$c_{xy}=\frac{\ell_{0}}{4}\cdot \left(\frac{1}{\ell_{1}}+\frac{1}{\ell_{2}}+\frac{1}{\ell_{3}}+\frac{1}{\ell_{4}}\right)-1=0.00883\;$$



(28)
$$\mathrm{with}\;u\left(c_{xy}\right)=\frac{\sqrt{u^{2}\left(\ell_{0}\right)+\left[u^{2}\left(\ell_{1}\right)+u^{2}\left(\ell_{2}\right)+u^{2}\left(\ell_{3}\right)+u^{2}\left(\ell_{4}\right)\right]/16}}{\ell_{0}}=0.00050$$



(29)
$$a=\frac{\ell_{0}}{2}\cdot \left(\frac{1}{\ell_{1}}-\frac{1}{\ell_{2}}\right)=-0.000040$$



(30)
$$\mathrm{with}\;u\left(a\right)=\frac{\sqrt{u^{2}\left(\ell_{1}\right)+u^{2}\left(\ell_{2}\right)}}{2\ell_{0}}=0.000036$$



(31)
$$\theta =\ell_{0}\cdot \left(\frac{1}{\ell_{3}}-\frac{1}{\ell_{4}}\right)=-0.000798$$



(32)
$$\mathrm{with}\;u\left(\theta \right)=\frac{\sqrt{u^{2}\left(\ell_{3}\right)+u^{2}\left(\ell_{4}\right)}}{\ell_{0}}=0.000074$$


All three parameters are dimensionless.

Observing the non-linearity RMS values in Table 3, the overall standard uncertainty estimation for non-linearity in the XY-plane was 
$u_{\mathrm{NL},xy}=0.7$
 µm.

### 3.3. Z-Axis Calibration

Figure 14 shows an example of a measurement of the spherical cap of a stainless steel reference sphere with a 4-mm nominal diameter (see Figure 9). It is a three-dimensional reconstruction of the sphere surface.

Using this information, the confocal microscope software can perform a least-square fitting to a spherical surface from which we could estimate the diameter of the sphere and RMS error of the fit. If the confocal software does not permit fitting a spherical surface, a code similar to the one described and listed in Appendix A can be used.

In this calibration, two different types of illumination were used (white and blue light), and measurements were taken in three orientations: 0°, 45°, and 90°. Finally, there were 
n=6
 measurements. Results obtained during the measurement of the bearing steel sphere of Figure 9 are presented in Table 4.

The average value 
${\bar{D}}_{m}$
 of the six diameters 
$D_{m}$
 was 
${\bar{D}}_{m}$
 = 3.9712 mm, and the standard deviation 
$s(D_{m})$
 was 0.0096 mm. We estimated 
$u({\bar{D}}_{m})$
 as:
(33)
$$u({\bar{D}}_{m})=\frac{s(D_{m})}{\sqrt{n}}\;=\;0.0039\;\mathrm{mm}.$$


The RMS error is an estimation of the repeatability in the Z-axis, which probably includes the non-linearity in the Z-axis. The mean value for this Z-axis repeatability was 
$s_{r}\left(z\right)$
 = 0.8 µm, which seems to be a reasonable value when compared with the Z-axis axial step of 2 µm.

The certified diameter 
$D_{0}$
 of the reference sphere was 
$D_{0}$
 = 4.0011 mm with a standard uncertainty 
$u(D_{0})$
 = 0.25 µm.

Using the expression of Section 2.3, the Z-axis calibration parameter 
$c_{z}$
 can be estimated as follows:
(34)
$$c_{z}=\frac{D_{m}}{D_{0}}\cdot \left(1+2c_{xy}\right)-1=0.0101$$



(35)
$$\mathrm{with}\;u(c_{z})=\sqrt{\frac{u^{2}(D_{0})+u^{2}({\bar{D}}_{m})}{D_{0}^{2}}+4u^{2}\left(c_{xy}\right)}=0.0014$$


The correlation coefficient 
$r\left(c_{z},c_{xy}\right)$
 was


(36)
$$r\left(c_{z},c_{xy}\right)=2\cdot \frac{u\left(c_{xy}\right)}{u\left(c_{z}\right)}=0.72$$


This correlation coefficient is clearly higher than zero, showing a strong positive correlation between 
$c_{z}$
 and 
$c_{xy}$
 that should be taken into account after calibration when needed. Correlation coefficients 
$r\left(c_{z},a\right)$
 and 
$r\left(c_{z},\theta \right)$
 are usually very small (lower than 0.01); therefore, correlation between 
$c_{z}$
 and parameters 
a
 and 
θ
 could be neglected.

### 3.4. Calibration for Roughness Measurements

As an example of data acquisition results when measuring a material roughness standard, the following figures show three-dimensional reconstructions of the surface of an aperiodic, metallic roughness standard (Figure 15) and a periodic, glass roughness standard (Figure 16).

Calibration was performed by repeating the measurements of six roughness standards 10 times (five zones, two orientations; see Table 1). The results are summarized in Table 5, showing average results 
$\bar{R}$
 of the 10 repeated measurements and corresponding standard deviations 
s(R)
. Direct readings 
R
 provided by the confocal microscope were obtained prior to introducing the calibration parameter 
$c_{z}$
 using only one sampling length 
$l_{r}=0.8$
 mm.

Therefore, these readings should be corrected by applying the following expression to take into account the Z-axis calibration:
(37)
$$R_{\mathrm{corrected}}=\bar{R}\cdot \left(1+c_{z}\right).$$


The authors followed the recommendations of ISO 4288 [44] that, for 
$0.1\;<R_{a}\leq 2\;$
mm, recommend five sampling lengths 
$l_{r}=0.8\;$
mm for a total evaluation length of 
$l_{n}=4\;\mathrm{mm}$
. Due to the limitations of the instrument’s field of view (see Section 2), only one sampling length 
$l_{r}=0.8$
 mm could be used. This reduction in the number of sampling lengths from five to one caused slightly lower values for 
$R_{a}\;$
and higher variabilities [45].

It can be concluded from Table 5 that a typical value for 
s(R)
 was 
s(R)=0.07
 μm, which was the quadratic average of repeatabilities of the first five standards. 

Note that corrected values 
$\bar{R}\cdot \left(1+c_{z}\right)$
 were always higher than the certified values 
$R_{0}$
 (compare the results from Table 5 to those from Table 1). It seems that for surface roughness similar to the nominal voxel height (
$w_{z}=2$
 μm), readings provided by the confocal microscope presented a positive bias caused by noise observed, for example, when measuring an optical flat (see Section 3.1, Figure 12). The RMS flatness observed when measuring the optical flat (0.54 µm) was slightly higher than the 
$R_{a}$
 values that were observed when measuring roughness standard #1, which is a quasi-flat surface (certified value 
$R_{a}=0.183$
 μm) for an instrument with a voxel height of 
2
 μm. The definition of 
$R_{a}$
 is similar but not equal to the definition of RMS flatness, but most importantly, 
$R_{a}$
 was evaluated after filtering the readings using a low-pass filter (defined through the sampling length
$\;l_{r}$
).

We suggest estimating the positive bias at each calibration point using the following expression, where 
$R_{0}$
 is the 
$R_{a}$
 certified value for the standard used at each calibration point:
(38)
$$b=\bar{R}\cdot \left(1+c_{z}\right)-R_{0}.$$


Its corresponding standard uncertainty 
u(b)
 is:
(39)
$$u\left(b\right)=\sqrt{u^{2}\left(R_{0}\right)+{\bar{R}}^{2}\cdot u^{2}\left(c_{z}\right)+\frac{s^{2}\left(R\right)}{n}}.$$


Using this approach, the calibration results are those values, 
$b_{i}$
 and 
$u\left(b_{i}\right),$
 which are presented in the two columns on the right side of Table 5. Index 
i
 refers to the roughness standard used. These results are represented graphically in Figure 17 in order to analyze their metrological compatibility. Red lines represent values corresponding to metallic, aperiodic standards #1, #2, and #3. Green lines represent values corresponding to standards #4 and #5. The blue line is the result from standard #6 that will not be taken into account. Vertical lines represents uncertainty intervals 
$b_{i}\;\pm \;U\left(b_{i}\right)$
 where the expanded uncertainties 
$U\left(b_{i}\right)=k\cdot u\left(\bar{b}\right)$
 were evaluated for a coverage factor
 k=2
 (see Section 6.2.1 in [24] for definitions of coverage factor and expanded uncertainty). When analyzing the compatibility between measurement results, it is very common to use a coverage factor of 
k=2
 to estimate the expanded uncertainties. The horizontal black solid line in Figure 17 corresponds to the average value 
$\bar{b}$
 of the first 
N=5
 roughness standards:
(40)
$$\bar{b}=\frac{\sum_{i=1}^{N}b_{i}}{N}=\frac{b_{1}+b_{2}+b_{3}+b_{4}+b_{5}}{5}=0.11\;\mu m$$


In order to make a correct estimation of average bias 
$\bar{b}$
, the correlation between bias
$\;b_{i},\;b_{j}$
 at each calibration points should be taken into account for the following reasons:Dominant contributions to uncertainties 
$u(b_{i})$
 are the calibration uncertainties 
$u(R_{0})$
 of the roughness standards.There is a high probability that all roughness standards were calibrated in the same calibration laboratory. Therefore, there will be strong correlation between them. 

There will be a high correlation between bias 
$b_{i}$
. We performed estimations in different situations, and it is possible to see correlation coefficients 
$r\left(b_{i},b_{j}\right)$
 as high as +0.8.

In order to simplify calculations, we suggest assuming
$\;r\left(b_{i},b_{j}\right)=+1$
, which leads to higher estimations for the uncertainty 
$u\left(\bar{b}\right)$
 of 
$\bar{b}$
. Then, it can be demonstrated that 
$u\left(\bar{b}\right)$
 was:
(41)
$$u\left(\bar{b}\right)=\frac{1}{N}\sum_{i=1}^{N}u\left(b_{i}\right)=\frac{u\left(b_{1}\right)+u\left(b_{2}\right)+u(b_{3})+u\left(b_{4}\right)+u(b_{5})}{5}=0.05\;µm$$


In Figure 17, the uncertainty interval 
$\bar{b}\pm U\left(\bar{b}\right)$
 is represented by the space between the higher and lower black dotted lines. 
$U\left(\bar{b}\right)=k\cdot u\left(\bar{b}\right)$
 is the expanded uncertainty of 
$\bar{b}\;$
evaluated with a coverage factor 
k=2
. Please note that all the uncertainty intervals 
$b_{i}\pm U\left(b_{i}\right)$
 overlap the interval 
$\bar{b}\pm U\left(\bar{b}\right)$
. Notwithstanding, point 
$b_{1}$
 is outside the interval 
$\bar{b}\pm U\left(\bar{b}\right)$
. This could indicate that some variability of the bias 
b
 was not taken into account in
$\;u\left(\bar{b}\right)$
. Therefore, a conservative approach would be to assume that there is a variability represented by 
δb
 that should be added to
$\;u\left(\bar{b}\right)$
. Suppose that 
δb
 is a uniform random variable of null mean and a full range
$\;b_{\max}-b_{\min}$
. Then, its standard uncertainty would be:
(42)
$$u\left(\delta b\right)=\frac{b_{\max}-b_{\min}}{\sqrt{12}}=0.06\;\mu m$$


In order to estimate the noise of the instrument, according to [40,41,42], we repeated 10 measurements over an optical flat (that of Figure 4) in two orientations: 0° and 90°. For a confocal microscope, an optical flat is a specimen with null roughness (very small in comparison with its noise). Therefore, values of 
$R_{a}$
 obtained over an optical flat are a very good estimation of the instrument noise. The average value and the standard deviation of the 10 
$R_{a}$
 values were


(43)
$${\bar{R}}_{a}=0.09\;\mu m$$



(44)
$$s(R_{a})=0.003\;\mu m$$


Therefore, a good estimation for the uncertainty component associated with noise instrument is


(45)
$$u_{\mathrm{noise}}={\bar{R}}_{a}=0.09\;\mu m$$


## 4. Discussion

Table 6 summarizes the results obtained during the confocal microscope calibration (Section 2).

Note that these results are only valid for measurements made with the same objective (10×). If other objectives are used, a whole recalibration is needed for each new objective.

The effects and their uncertainties were the highest for parameters 
$c_{xy}$
 and 
$c_{z}$
. If their effects were not corrected, their contribution to the relative expanded uncertainty would be around 1%.

Fortunately, the software of confocal microscopes usually permits users to introduce their value in order to compensate their effects. If this compensation is done, their contribution to the relative expanded uncertainty is reduced to 0.3%. 

The effect of parameter 
a
 (difference between pixel lengths along X and Y axes) was negligible. Its absolute value was lower than its expanded uncertainty 
U(a)=k·u(a)
 (for 
k=2
); therefore, the null hypothesis 
a=0
 could not be rejected. Its contribution to the relative expanded uncertainty was very low (around 0.01%). 

The effect of parameter 
θ
 (perpendicularity error between X and Y axes) seemed to be significant (its absolute value was clearly higher than its expanded uncertainty), but its contribution to the relative expanded uncertainty (around 0.1%) was clearly negligible in comparison with 
$c_{xy}$
 and 
$c_{z}$
.

The contributions of XY plane RMS flatness (
$u_{\mathrm{FLT}})$
 and the non-linearity in X and Y axes were clearly lower than the voxel dimensions (
$w_{xy}=1.65$
 μm and 
$w_{z}=2$
 μm). Therefore, the instrument adjustment performed by the manufacturer seems to have been good.

Repeatabilities in the XY plane and in the Z-axis, in comparison with the voxels dimensions, were low. Again, this can be used to conclude that the instrument was working well.

In roughness measurements (which only apply when using the 
$R_{a}$
 parameter), the repeatability
 s(R)
, average bias 
$\bar{\;b}$
, bias variability
 u(δb
), and instrument noise 
$u_{\mathrm{noise}}$
 were very small in comparison with voxel height
$\;w_{z}=2$
 μm.

### 4.1. Expanded Uncertainty Estimation for Length Measurements in the XY Plane

As was pointed out, the instrument software usually permits users to introduce parameters 
$c_{xy}$
 and 
$c_{z}$
 in order to apply the corresponding corrections. On the contrary, parameters 
a
 and 
θ
 cannot be introduced. Therefore, the effect of uncorrected, non-null parameters 
a
 and 
θ
 would be taken into account as a systematic effect whose equivalent standard uncertainties would respectively be 
$\left|a\right|/\sqrt{3}$
 and 
$\left|\theta \right|/\sqrt{3}$
. We supposed that it was equivalent to the introduction of two components, 
δa
 and 
δθ
, uniformly distributed along 
[−a.+a]
 and 
[−θ.+θa],
 respectively.

For length measurements performed in the XY plane, the following expression could be a good estimate of its expanded uncertainty, where the pixel width component was estimated as 
$w_{xy}/\sqrt{12}$
 (uniformly distributed between 
$\pm w_{xy}/2$
):
(46)
$$U\left(L_{xy}\right)=k\cdot \sqrt{L_{xy}^{2}\cdot \left\{u^{2}\left(c_{xy}\right)+\frac{a^{2}}{3}+u^{2}\left(a\right)+\frac{1}{2}\left[\frac{\theta^{2}}{3}+u^{2}\left(\theta \right)\right]\right\}+u_{\mathrm{NL},xy}^{2}+s_{r}^{2}\left(x\right)+\frac{w_{xy}^{2}}{12}}\leq \;\leq 1.9\;\mu m+\frac{L}{1600}.$$


Uncertainty components for which no distribution was described were supposed to have normal distributions. In such situations, where there are many uncertainty components (eight components) and most of them are normally distributed and they contribute similarly to the total combined uncertainty, it can be supposed that the output variable 
$L_{xy}$
 is normally distributed [25]. Therefore, a coverage factor 
k=2
 can be used when computing the expanded uncertainty, assuming a coverage probability of approximately 95%.

### 4.2. Expanded Uncertainty Estimation for Height Measurements along the Z-Axis

For height measurement (
0≤h≤100 
μm—the Z range approximately covered by the sphere cap measured), the following expression gives us a reasonable estimation of its expanded uncertainty 
U(h)
 for a coverage factor 
k=2
:
(47)
$$U\left(h\right)=k\cdot \sqrt{h^{2}\cdot u^{2}\left(c_{z}\right)+u_{\mathrm{FLT}}^{2}+s_{r}^{2}\left(z\right)+\frac{w_{z}^{2}}{12}}\leq 2.2\;\mu m+\frac{h}{120}$$


Now there are four uncertainty components, where three are distributed normally, and only one 
$w_{z}$
 is distributed uniformly, but 
$w_{z}$
 is never the dominant contribution. Again, in a situation such as this, a coverage factor 
k=2
 can be used when computing the expanded uncertainty, assuming a coverage probability of approximately 95% [25]. 

### 4.3. Expanded Uncertainty for Roughness Measurements

Following the recommendations of DKD-R 4-2 [40,41,42], a model for a corrected 
$R_{a}$
 roughness measurement performed after instrument calibration would be:
(48)
$$R_{a}=\bar{R}\cdot \left(1+c_{z}\right)-\left(\bar{b}+\delta b\right)+\delta R_{\mathrm{noise}}$$

where now 
$\bar{R}$
 is the average of 
m
 repeated measurements made over the specimen being measured, and 
$\delta R_{\mathrm{noise}}$
 is a random variable of null mean distributed normally with standard deviation 
$u_{\mathrm{noise}}$
. The standard uncertainty of 
$R_{a}$
 is


(49)
$$u(R_{a})=\sqrt{\frac{s^{2}\left(R\right)}{m}+{\bar{R}}^{2}\cdot u^{2}\left(c_{z}\right)+u^{2}\left(\bar{b}\right)+u^{2}\left(\delta b\right)+u_{\mathrm{noise}}^{2}}$$


The expanded uncertainty 
$U(R_{a})$
, using a coverage factor 
k
 is


(50)
$$U(R_{a})=k\cdot \sqrt{\frac{s^{2}\left(R\right)}{m}+{\bar{R}}^{2}\cdot u^{2}\left(c_{z}\right)+u^{2}\left(\bar{b}\right)+u^{2}\left(\delta b\right)+u_{\mathrm{noise}}^{2}}$$


In this case, there are five uncertainty components, 
δb
 is distributed uniformly, and 
$\bar{R}$
 follows a *t*-Student distribution with 
ν=m−1
 degrees of freedom. If we compute the degrees of freedom of the output variable 
$R_{a}$
, the result is approximately 
$\nu \left(R_{a}\right)=\;$
 30. With 
$\nu \left(R_{a}\right)>10$
, it is possible to use a coverage factor 
k=2
 corresponding to a coverage probability of approximately 95% [25]. Then, assuming that measurements will be repeated 
m=3
 times, the expanded uncertainty would be

(51)
$$U(R_{a})<0.25\;\mu m$$


This value is very good for an instrument with a voxel height of 
$w_{z}=2$
 μm.

### 4.4. Propagation of Uncertainty When Measuring the Radius of a Cylindrical Surface

As an example of uncertainty propagation in dimensional measurements not directly covered in Section 4.2 and Section 4.3, we present the case of a measurement of the radius of a cylindrical surface (see Figure 18) of a steel bar with a nominal value of 2.75 mm.

The uncertainty propagation was done using Monte Carlo simulation [46]. The model function used during the simulation of the coordinates of the surface’s points was the following:
(52)
$$\left[\begin{array}{c} x\\ y\\ z \end{array}\right]=\left[\begin{array}{ccc} 1+c_{xy}+a+\delta a & \left(\theta +\delta \theta \right)/2 & 0\\ \left(\theta +\delta \theta \right)/2 & 1+c_{xy}-a-\delta a & 0\\ 0 & 0 & 1+c_{z} \end{array}\right]\cdot \left[\begin{array}{c} p\\ q\\ r \end{array}\right]+\left[\begin{array}{c} \delta x\\ \delta y\\ \delta z \end{array}\right]$$

where 
(p,q,r)
 are the coordinates provided by the confocal microscope during the measurement. They were not simulated. Table 7 enumerates variables that were simulated and how the simulation was performed. Corrections 
a
 and 
t
 were not applied, because the instrument software cannot take them into account. In order to take into account the effect of not applying these corrections, 
δa
 and 
δt
 were introduced, and 
a
 and 
t
 are supposed to be normal distributions with zero mean (not applying corrections) and typical uncertainty 
u(a)
 and 
u(t)
. 
δa
 and 
δt
 have uniform distribution with zero mean and typical uncertainties 
$\left|a\right|/\sqrt{3}\;$
and 
$\left|\theta \right|/\sqrt{3}$
. 
δp,  δq, and δr
 represent the repeatability effects over coordinates 
x,y,  and z
. We supposed that they had normal distributions with zero mean and typical uncertainties 
$u\left(\delta p\right)=\left(\delta q\right)=s_{r}\left(x\right)=s_{r}\left(y\right)$
 and 
$u\left(\delta r\right)=s_{LS}=1.0$
 μm, where 
$s_{LS}$
 is the root mean squared error observed when fitting a cylindrical surface to measured points 
(x,y,z)
 using a least-squares fit.

A total of 
$N=10^{4}\;$
simulations were generated. For each simulation, a value 
$R_{i}$
 was obtained for the radius of the cylindrical surface. Figure 19 shows the histogram of the simulated radius of the cylindrical surface. The red smooth line represents the best approximation of the histogram through a normal distribution. Differences between the red line and the histogram were small enough to be negligible. It is likely that upon increasing the number of simulations (the advisable value for *N* when there is no time limit during the execution to the simulation process is 
$N=10^{6}$
 [46]), these differences would be smaller. For this reason, a coverage factor 
k=2
 (corresponding to an approximate coverage probability of 95%) was chosen in previous sections: final distributions are usually very close to normal distribution where 
k=2
 corresponds to a coverage probability of 95.45%.

The final result was 
R=(2.7907±0.0061) mm
, where the expanded uncertainty 
U(R)=k·s(R)
, where now 
s(R)
 is the standard deviation of 
N
 simulated values 
$R_{i}$
. A coverage factor 
k=2
 for a coverage probability of approximately 95% was used.

A similar approach could be used to propagate uncertainties when using the confocal microscope for other types of dimensional or angular measurements. 

## 5. Conclusions

A complete calibration procedure that provides adequate traceability to confocal microscopes used in submillimeter coordinate metrology was presented. This procedure provides adequate traceability for length and roughness measurements performed with confocal microscopes and can be easily adapted to calibrate other 3D optical instruments (e.g., focus variation microscopes). The calibration procedure is as simple as possible, as it was designed to be implemented in industrial environments. Reference material standards were chosen to be easy to find and easy to calibrate again in industrial environments. The calibration procedure covers all the key points of operation of a confocal microscope. It permits the estimation of:Amplification coefficients 
$\alpha_{x}=1+c_{xy}+a$
, 
$\alpha_{y}=1+c_{xy}-a$
, and 
$\alpha_{z}=1+c_{z}$
.Non-linearity errors.Perpendicularity error 
θ
 between X and Y axes.Relative difference 
2a
 in pixel dimensions along X and Y axes.Repeatabilities when measuring lengths or heights.Flatness deviations in the XY plane.Bias deviation 
b
 when measuring roughness. Instrument noise when measuring roughness.Repeatability when measuring roughness.

Some of these parameters (amplification coefficients, flatness deviation in XY plane) can usually be introduced in the instrument software to compensate for their effects. Others cannot be compensated (i.e., 
θ,a
) but if high values are detected, the user can ask the instrument manufacturer to adjust and/or repair the instrument to reduce their effects. Even if they are not introduced, an alternative approach is presented here to account for the fact that these corrections were not applied.

Uncertainty estimations were carried out for all parameters following the mainstream GUM method. In addition, for measurements of lengths and roughness, expressions for expanded uncertainties of measurement carried out by the instrument were provided. There are other types of measurements, such as angular measurements, that were not addressed in this paper due to limitations in the extent of the text. Notwithstanding, all the information needed to propagate uncertainties to these other types of measurements is provided in the paper, as can be demonstrated through an example solved using Monte Carlo simulation.

The procedure described in this paper can be easily adapted to calibrate other optical instruments in the submillimeter range, which are capable of providing 3D information of surfaces being observed by them (e.g., focus variation microscopes). For example, the material of some of the reference material standards would have to be changed, but the core of the procedure would remain the same.

## Figures and Tables

**Figure 1 materials-12-04137-f001:**
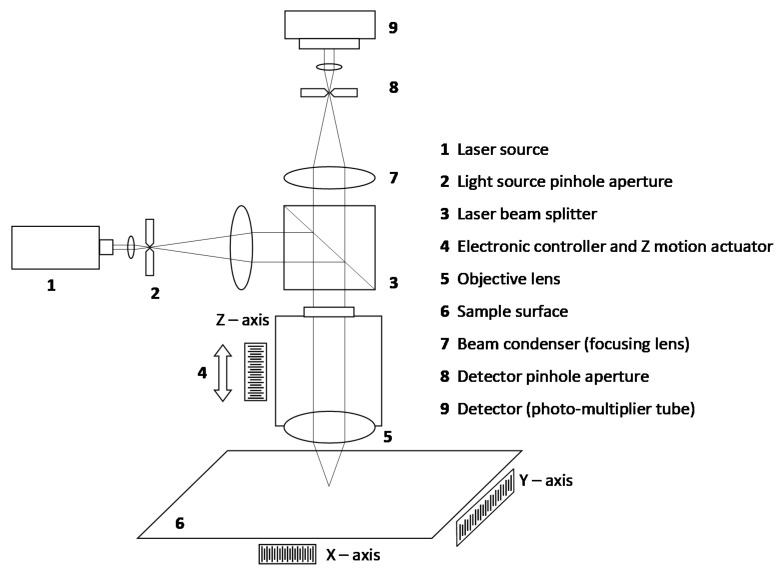
Scheme of a confocal microscope [5,18,22].

**Figure 2 materials-12-04137-f002:**
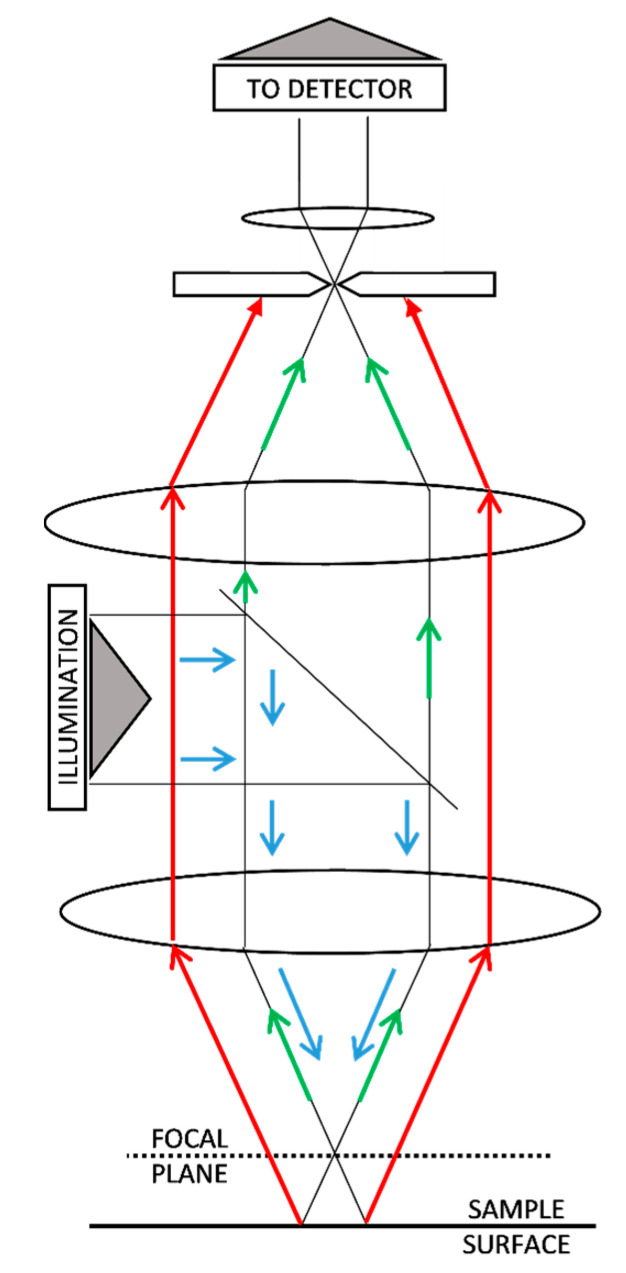
Filtration of out-of-focus signal (red) in confocal microscopy [5,7,16,17,21] in comparison with in-focus signal (green).

**Figure 3 materials-12-04137-f003:**
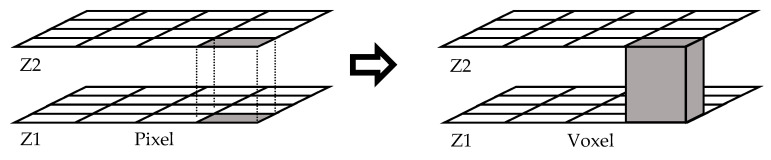
Transformation from pixel to voxel.

**Figure 4 materials-12-04137-f004:**
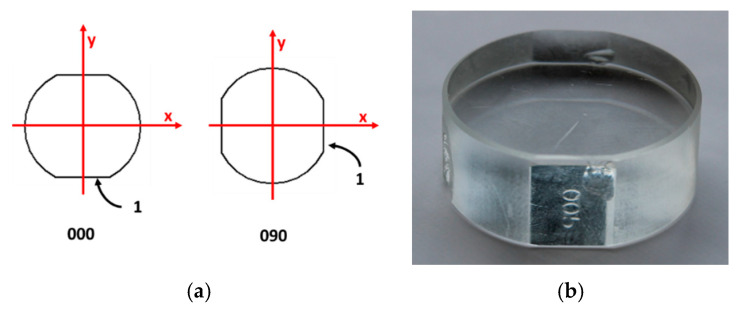
(**a**) Positions of the optical flat during calibration; (**b**) optical flat.

**Figure 5 materials-12-04137-f005:**
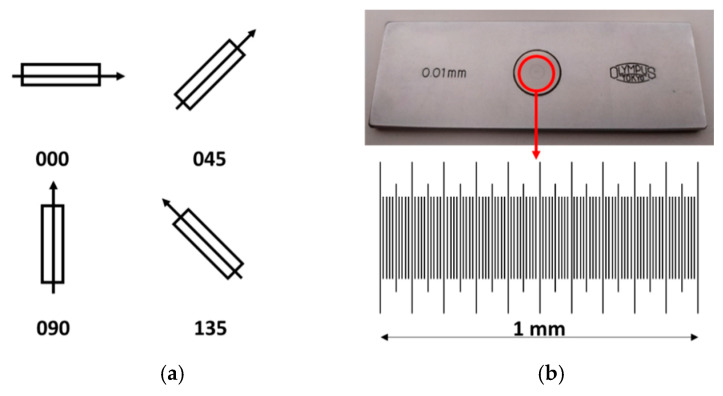
(**a**) The different positions of scanning for the stage micrometer; (**b**) the stage micrometer used during calibration.

**Figure 6 materials-12-04137-f006:**
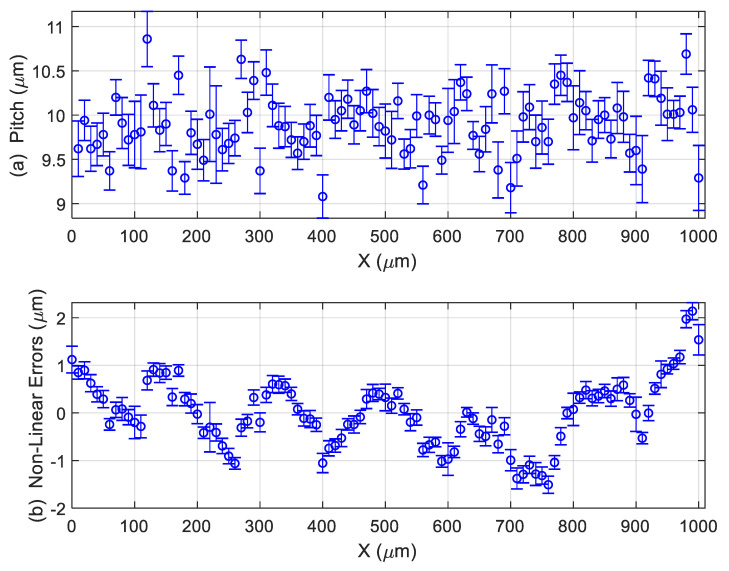
Measurement of stage micrometer in position 0°: (**a**) pitch measurements results in µm; (**b**) non-linear errors in µm.

**Figure 7 materials-12-04137-f007:**
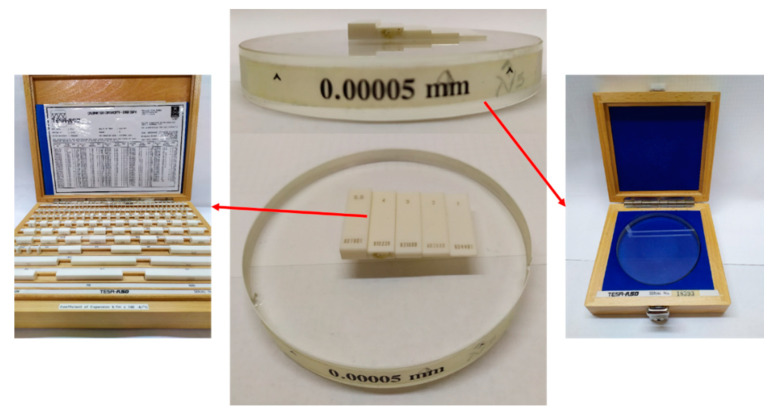
A step gauge built on an optical flat (with total flatness error 0.00005 mm) and a set of gauge blocks.

**Figure 8 materials-12-04137-f008:**
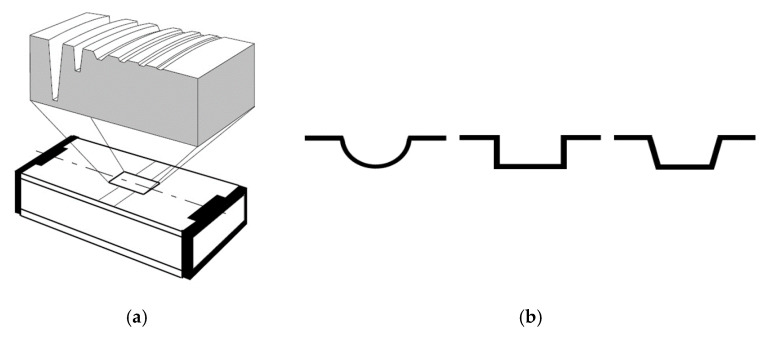
This figure shows (**a**) a typical model of a step height standard; and (**b**) different models of step height standards’ grooves (ISO 5436-1 types A and B) [30,35,36,37].

**Figure 9 materials-12-04137-f009:**
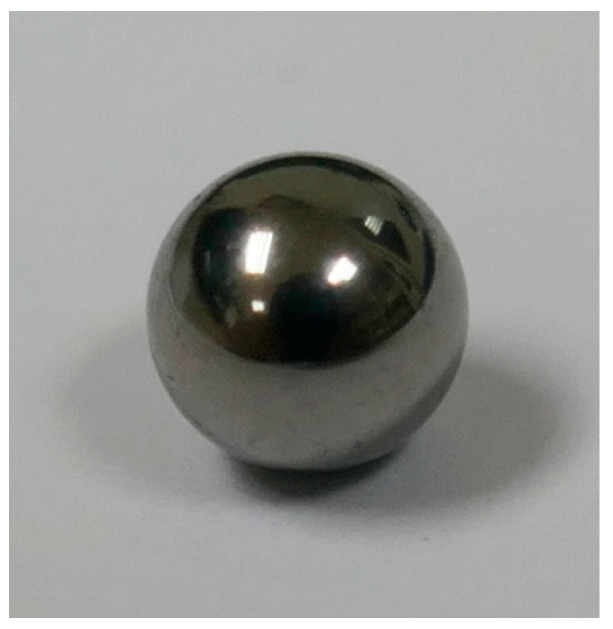
Steel sphere used in calibration.

**Figure 10 materials-12-04137-f010:**
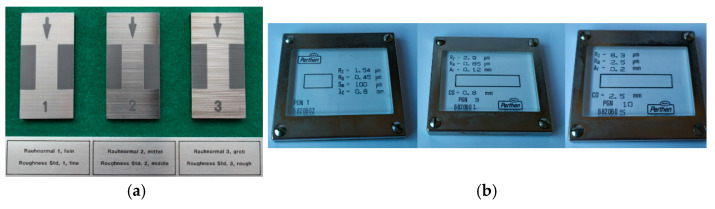
Step height standards used during calibration: (**a**) aperiodic, metallic standards and (**b**) periodic, glass standards.

**Figure 11 materials-12-04137-f011:**
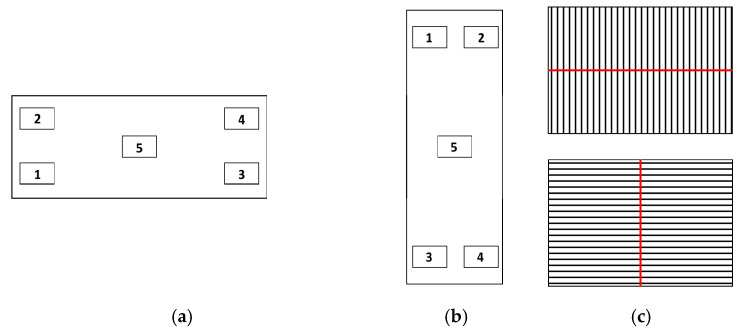
Location of the five scanning positions for roughness calibration: (**a**) horizontal orientation; (**b**) vertical orientation; and (**c**) location of measurement lines.

**Figure 12 materials-12-04137-f012:**
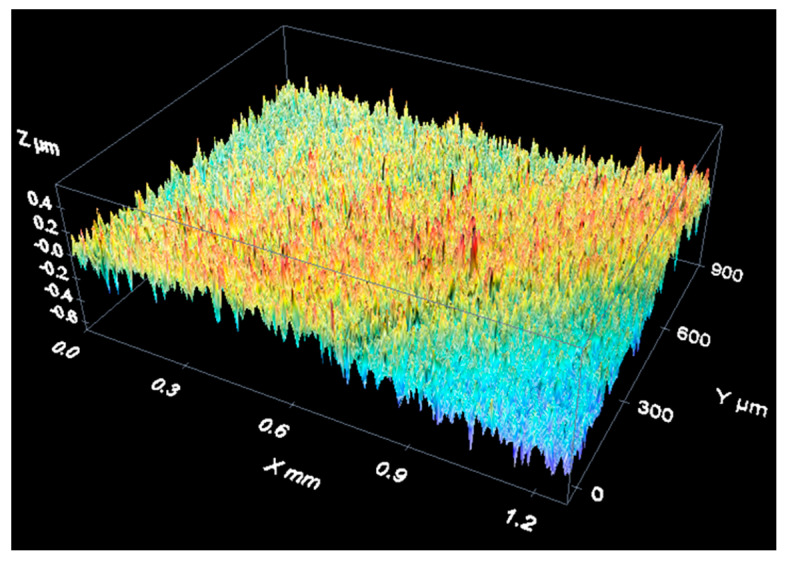
Result of the flatness measurement performed over the optical flat.

**Figure 13 materials-12-04137-f013:**
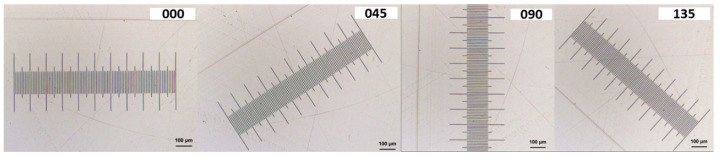
Different measurement positions (0°, 45°, 90°, 135°) of the stage micrometer.

**Figure 14 materials-12-04137-f014:**
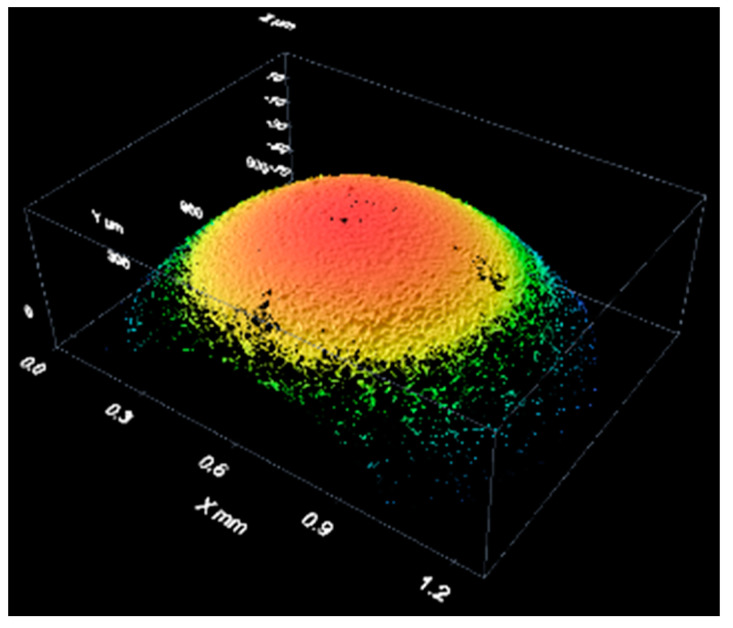
Results of the measurement of a bearing sphere with white light.

**Figure 15 materials-12-04137-f015:**
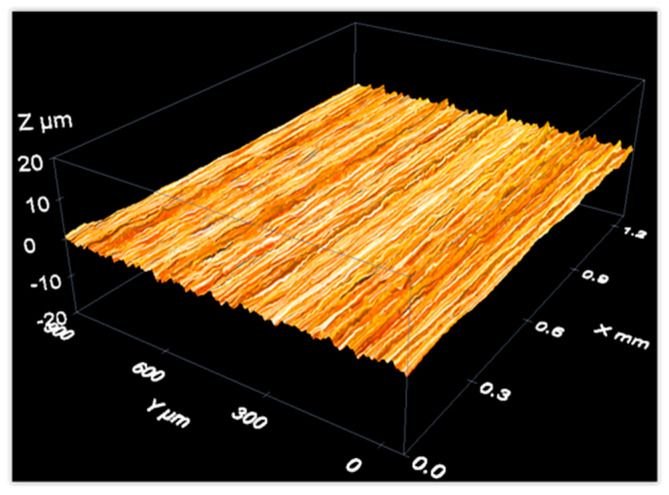
Measurement of an aperiodic roughness standard with a confocal microscope: 3D view of the measurement results.

**Figure 16 materials-12-04137-f016:**
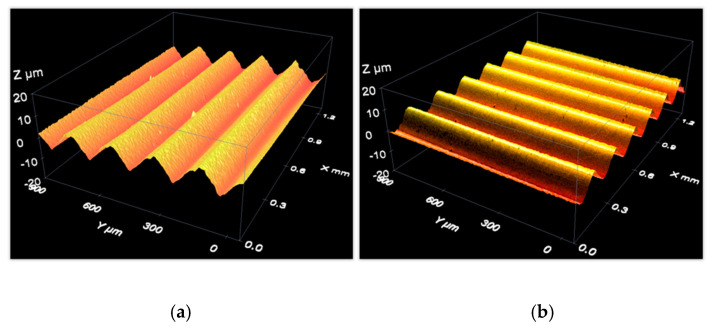
Measurement of a periodic roughness standard with a confocal microscope: 3D view of the measurement results with roughness lines (**a**) parallel to the X-axis and (**b**) parallel to the Y-axis.

**Figure 17 materials-12-04137-f017:**
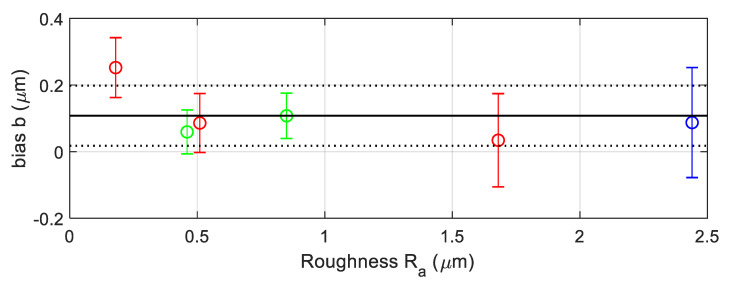
Bias observed at each calibration point (roughness calibration).

**Figure 18 materials-12-04137-f018:**
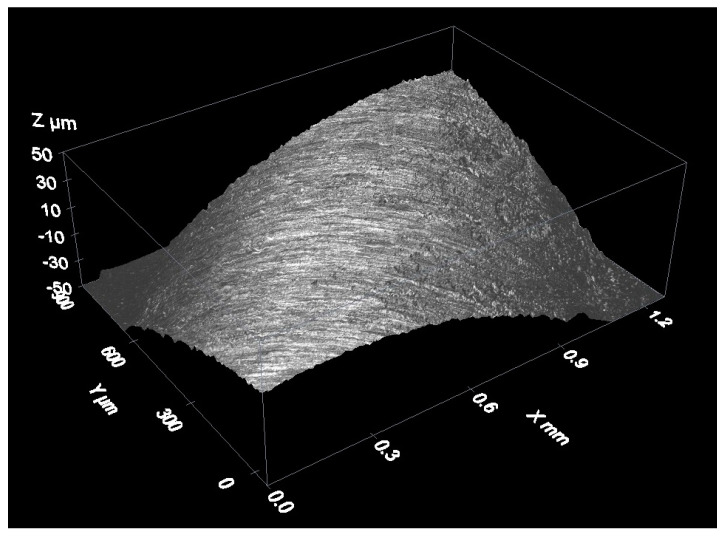
Measurement of a cylindrical surface.

**Figure 19 materials-12-04137-f019:**
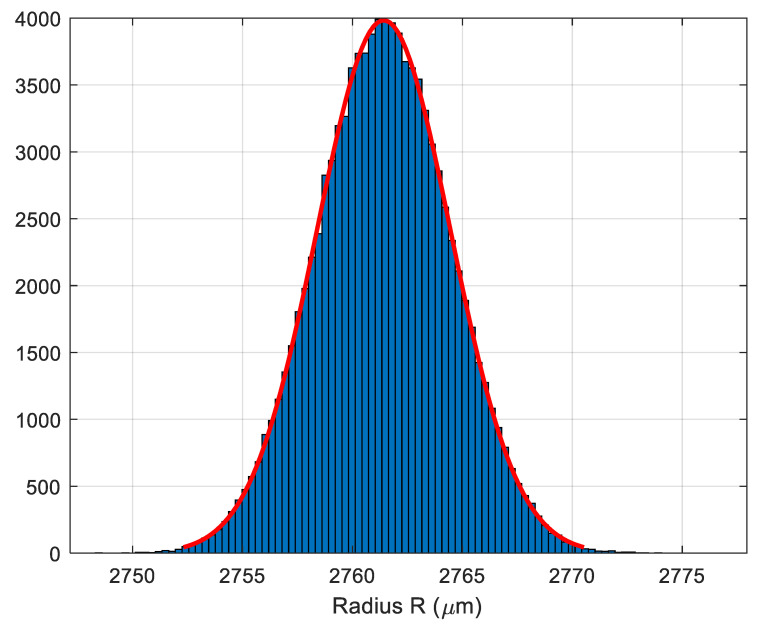
Histogram of the radius 
R
 of the cylindrical surface.

**Table 1 materials-12-04137-t001:** Nominal values and the uncertainties of the material reference standards used during calibration.^1^

$S_{m}$
 is a spacing parameter defined as the mean spacing between peaks. 
$S_{m}$
 values included in this table are only informative. RMS: root mean square error.

Reference Measurement Std.	Parameter	Certified Value (µm)	Std. Uncertainty (k=1) (μm)
Optical flat	Total flatness error	0.118	0.025
	RMS flatness	0.028	0.007
Stage Micrometer	Average pitch $\ell_{0}$	9.980	0.005
Sphere	Diameter $D_{o}$	4 001.08	0.25
Roughness std. #1 metallic, aperiodic	$R_{a}$ ( $R_{0}$ ) $S_{m}$ ^1^	0.183 48	0.039
Roughness std. #2 metallic, aperiodic	$R_{a}$ ( $R_{0}$ ) $S_{m}$ ^1^	0.512 185	0.041
Roughness std. #3 metallic, aperiodic	$R_{a}$ ( $R_{0}$ ) $S_{m}$ ^1^	1.677 176	0.057
Roughness std. #4 glass, periodic	$R_{a}$ ( $R_{0}$ ) $S_{m}$ ^1^	0.460 100	0.030
Roughness std. #5 metallic, aperiodic	$R_{a}$ ( $R_{0}$ ) $S_{m}$ ^1^	0.850 120	0.030
Roughness std. #6 glass, periodic	$R_{a}$ ( $R_{0}$ ) $S_{m}$ ^1^	2.440 200	0.080

**Table 2 materials-12-04137-t002:** RMS flatness measured with the confocal microscope in positions 0° and 90°.

Position	RMS Flatness (μm)
0°	0.48
90°	0.59

**Table 3 materials-12-04137-t003:** Measurements of the average pitch 
$\ell_{i}$
 in different orientations.

Position		$\mathrm{Average}\;\mathrm{Pitch}\;\ell_{i}\;(\mu m)$	$\mathrm{Uncertainty}\;u(\ell_{i})\;(\mu m)$	Repeatability s (μm)	Non-Linearity RMS (μm)
1	0°	9.892 34	0.000 53	0.34	0.71
2	45°	9.897 33	0.000 57	0.38	0.60
3	90°	9.891 56	0.000 49	0.42	0.69
4	135°	9.889 50	0.000 47	0.34	0.61

**Table 4 materials-12-04137-t004:** Root mean square error and diameter 
$D_{m}$
 of the spherical caps fitted using least squares.

Position	Illumination	RMS Error (μm)	$\mathrm{Diameter}\;D_{m}\;(\mathrm{mm})$
0°	Blue	0.86	3.9740
45°	Blue	1.08	3.9562
90°	Blue	1.08	3.9638
0°	White	0.86	3.9740
45°	White	0.89	3.9828
90°	White	0.87	3.9766

**Table 5 materials-12-04137-t005:** Results obtained when calibrating the confocal microscope described in Section 2 using six roughness standards (Table 1).

Reference Meas. Std.	Average *R_a_* $\bar{R}$ (μm)	Repeatability s(R) (μm)	Corrected *R_a_* $\bar{R}\cdot \left(1+c_{z}\right)\;(\mu m)$	Bias Estimation b (μm)	Standard Uncertainty u(b) (μm)
Roughness std. #1	0.43	0.06	0.43	0.25	0.04
Roughness std. #2	0.59	0.06	0.60	0.08	0.05
Roughness std. #3	1.70	0.11	1.71	0.04	0.07
Roughness std. #4	0.51	0.04	0.52	0.06	0.03
Roughness std. #5	0.95	0.05	0.96	0.11	0.03
Roughness std. #6 ^1^	2.50	0.06	2.53	0.09	0.08

^1^ Values obtained when measuring roughness standard #6 are included in this table only for informative reasons. Measurements of this standard were made using a sampling length 
$l_{r}=0.8$
 mm, because of the reduced field of view of the instrument, instead of a sampling length 
$l_{r}=2.5$
 mm, as recommended by ISO 4288 [44].

**Table 6 materials-12-04137-t006:** Results of calibration.

Parameter	Value	Units	Standard Uncertainty
$c_{xy}$	0.008 83	-	0.00050
a	−0.000040	-	0.000036
θ	−0.000798	-	0.000074
$c_{z}$	0.0101	-	0.0014
$r\left(c_{xy},c_{z}\right)$	0.72	-	-
$u_{\mathrm{FLT}}$	0.54	µm	-
$u_{\mathrm{NL},xy}$	0.70	µm	-
$s_{r}\left(x\right)=s_{r}\left(y\right)$	0.40	µm	-
$s_{r}\left(z\right)$	0.80 ^1^	µm	-
$\bar{b}$	0.11	µm	0.05
δb	0	µm	0.06
s(R)	0.07	µm	
$u_{\mathrm{noise}}$	0.09	µm	

^1^ Non-linearity in Z-axis is included in 
$s_{r}\left(z\right)$
.

**Table 7 materials-12-04137-t007:** Variables simulation.^1^ Corrections 
a
 and 
t
 were not applied.

Variable	Mean Value	Units	Standard Uncertainty	Distribution Type

$c_{xy}$	0.00883	-	0.00050	Normal
a	0 ^1^	-	0.000036	Normal
δa	0		$u\left(\delta a\right)=\frac{a}{\sqrt{3}}=0.000023$	Uniform
θ	0 ^1^	-	0.000074	Normal
δa	0		$u\left(\delta a\right)=\frac{a}{\sqrt{3}}=0.00046$	Uniform
$c_{z}$	0.0101	-	0.0014	Normal
$r\left(c_{xy},c_{z}\right)$	0.72	-	-	-

δp,δq	0	µm	$s_{r}\left(x\right)=s_{r}\left(y\right)=0.40$	Normal
δr	0	µm	$s_{LS}=1.0$	Normal

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
