# Peer review of "Industrial Calibration Procedure for Confocal Microscopes"

_materials, 2019, doi:10.3390/ma12244137_

Round 1

Reviewer 1 Report

Please state clearly what is a research issue in the paper. There is only a calibration procedure taken from ISO 12179 and used for a confocal microscope, so a standard issue. What scientific problem did the Authors intended to solve?

Please change the title. There is nothing in research that is related to 3D printing. A confocal microscope calibrated this way can be used for any purpose. So he whole introduction about 3D printing is not relevant.

Working principle of a confocal microscope is well known and described even in respective part of ISO 25178. There is no need to present it in details.

Some references are not appropriate. E.g. at figure 1 there are three references, 16 and 23 should be removed leaving only the last one. The same applies to 3D printing references.

Author Response

First of all, we would like to thank you for your work as a reviewer and for the comments and the feedback you have given us. They have allowed us to clearly improve our manuscript.

Probably, the way was the first version of our manuscript written could make the reader assume that the calibration procedure described in it, it is only valid for a 2D roughness measurement. And for this type of measurement, it is true that a valid calibration procedure it is described in ISO 12179.

But our real objective is not the 2D roughness measurements, is to carry angular and dimensional measurements. That is when the confocal microscope is used like a small Coordinate Measuring Machine (CMM). And, as a secondary objective, the 2D roughness measurements.

Anyway, our personal opinion is that before trying to achieve traceability for roughness measurements (2D or 3D) the traceability of coordinates (x,y,z) should be assured before. And this is the objective of our work. And this is an open problem as it can be very good described by R.Leach in a recent article:

https://www.researchgate.net/publication/336567430_Calibration_of_optical_surface_and_coordinate_measuring_instruments_Pleading_for_a_complete_framework

In order to make clearer that our principal objective is assuring the traceability in dimensional measurements (not roughness) an additional example has been introduced (section 4.4) where the measurement of cylindrical surface is analyzed.

We hope that in the new version of the manuscript, with the modification we have made in abstract, in the introduction and section 4.4, the objective (calibration of the confocal microscope as a small CMM) is clearer transmitted.

We agree with you that the calibration procedure we present can be applied to any type of confocal microscopes, no matters if it is used to work in Additive Manufacturing or not. Even more, it could be applied to other optical instruments as focus variation microscopes.

The reason why we wrote the first version of the manuscript with many references to Additive Manufacturing was, because of the confocal microscope we have been using is an instrument used to control processes of Additive Manufacturing and we were always thinking in this particular application. But this point of view was very reduced.

Because of that:

We propose a new title for the paper: “Industrial calibration procedure for Confocal Microscopes”, deleting any reference to “Additive Manufacturing” We have re-written the introduction We have modified the abstract.

It’s true that ISO 25718 included a good description of the working principle of a confocal microscope, but we consider that it will be useful for the reader to have a short description of this working principle in the body of the text, but, in the second version of the manuscript this description is shorter.

Following your comment, we have revised all cases where there are multiple references (as in figure 1) and we have reduced the number of references. All references (and text) corresponding to Additive Manufacturing have been deleted.

Sincerely yours,

Alberto Mínguez

Jesús de Vicente

Reviewer 2 Report

The authors present a simple method for the calibration of confocal microscopes that can easily be implemented in workshop or industrial environments. While the paper is well structured and relatively well written, the authors should address the following comments:

the title states "used in micro-additive manufacturing processes", however i would think that confocal microscopy can be used in various application domains. Is there a specific reason why additive manufacturing (AM) is chosen to be included in the title and in the introduction as a specific application domain? 
The way the title is formulated is a bit misleading, in the sense that it gives the impression that the confocal microscope is being used in-line in the AM process, but it is used off-line (to assess the parts after they have been printed, not during the printing process itself).  Apart from the title and the introduction, there is no mention of or link with AM further on in the manuscript. there are quite a lot of grammatical and spelling errors. It is recommended that the paper is reviewed by a native English speaker.
(e.g. line 141 "procedures described later ON"; line 158 "all uncertainties will BE investigated"; line 177 "the software of THE confocal MICROSCOPE provides"; line 231 "Document [42] proposeS calibrating... a step gauge builT with..."; etc.) line 122 There is something wrong with the sentence "In this way, to make an interpolation between consecutive images to create the computational model of the scanned surface is needed." What do the authors mean here? line 148: "the instrument is going to be used for single topography measurements" --> I would add "if" at the beginning of this sentence. Now it seems that one always uses it in this mode and that XY stage movements never contribute to the error. Line 178, 184 why do the authors talk about a "flatness defect"? They should also clarify why the RMS flatness 'defect' is more stable than the total flatness 'defect', or provide a reference. Line 212 the authors should provide some information on the "special software used for this task" (is it open source?). A better description is needed to explain how Figure 6a is obtained.  Line 217 and Figure 6b: what are the non-linear errors and how have they been calculated?  Figure 7: what is the 0.00005mm indication on the picture? what does it refer to? Line 249 "the software of the confocal microscope permits to fit a spherical surface" --> can this be generalized or is it only applicable for the specific confocal microscope used in this work? Maybe it would be good to point the readers to available open source code to do this fitting it their microscope software would not allow this fit. Eq. (20): should it not be z = r + (1+c_z)r? Line 268: the correlation coefficient r can easily be confused with the (p,q,r) reading of the microscope. Line 314: why would optical measurements tend to overestimate surface roughness over stylus profilometry? Line 326: It would be useful to explain the bandwidth matching concept in a few lines rather than just referring to [53]. Figure 12: while the authors state it shows the absence of visible curvature, it would seem that some cilinder can be observed in the measurement (the diagonal seems to be red colored and moving away from that diagonal in a perpendicular direction goes towards a blue color).  Table 2: again, the term "flatness defect" sounds strange Table 3: How are the values l_i and u(l_i) obtained exactly? And what is the "non linearity RMS" exactly? How is this defined? Line 469 and further: how is the coverage factor k defined and why is it set equal to 2? Can all conclusions made in the paper be generalized? E.g. line 550: "some of these parameters can be introduced in the instrument software" --> is that valid for the confocal microscope used in this paper or for all confocal microscopes on the market?

Author Response

First of all, we would like to thank you for your work as a reviewer and for the comments and the feedback you have given us. Some of them have been very useful to clarify some aspects of our work and make it applicable to a wider area.

We agree with you that the calibration procedure we present can be applied to any type of confocal microscope, no matters if it is used to work in Additive Manufacturing or not. Even more, it could be applied to other optical instruments as focus variation microscopes. Because of that:

We propose a new title for the paper: “Industrial calibration procedure for Confocal Microscopes”, deleting any reference to “Additive Manufacturing” We have re-written the introduction We have modified the abstract.

The reason why we wrote the first version of the manuscript with many references to Additive Manufacturing was that the confocal microscope we have been using is an instrument used to control processes of Additive Manufacturing and we was always thinking in this particular application. But this point of view was very reduced.

Following your recommendation, we have sent the manuscript to be reviewed by a native English speaker. Therefore, we hope all grammatical and spelling errors you detected in the first version of our manuscript have been corrected.

In the attached document we revised, point by point, the comments you made us and the modifications we have made in the second version of the manuscript to take into account your comments.

Sincerely yours,

Alberto Mínguez

Jesús de Vicente

Reviewer 3 Report

The paper is interesting, but the calibration steps are strongly related to the utilized confocal microscope and the relative software. In the reviewer’s opinion, the developed simple calibration method is difficult to apply in industrial environment.

However, minor revisions are required as suggested below:

Figure 1 and 2 are not necessary. The confocal microscope description and functioning are already reported in the body of the text

Diverse figures are not listed in the body of the text:

At line 123-124 of the pdf file, please cite Figure 3 At line 180-181 of the pdf file, at the end of the following sentence: “The calibration will be done in two positions (0° and 90°) and, therefore, two measurements will be obtained” cite “(Figure 4)” At line 188 of the pdf, at the end of the following sentence: “…..in four positions” cite “(Figure 5)” At line 231-232 of the pdf file, at the end of the sentence cite Figure 7 At line 236-240 of the pdf file, cite Figure 8 in the body of text. At line 246-247 of the pdf file, cite Figure 9 in the body of text.

Moreover, at line 363, in Figure 13, and in Table 3, substitute (0°, 90°, 45°, 135°) with (0°, 45°, 90°, 135°)

In Figure 6, please improve the readability of the text and figure.

Author Response

First of all, we would like to thank you for your work as a reviewer and for the comments and the feedback you have given us. They have allowed us to clearly improve our manuscript.

Probably, the way was written the first version of our manuscript could make the reader assume that the calibration procedure described in it, is only valid for a particular model of the confocal microscope as you pointed out. But we think that it is a procedure that can be applied to any type of confocal microscope. Even more, it could be applied to other optical instruments as focus variation microscopes. The reasons are the following:

In any type of measuring microscope is possible to measure distances between marks. Probably, sometimes, measurement has to be done manually instead of using a software that automatically detects all marks. But, it is only more laborious. Therefore, the calibration on the XY plane with the stage micrometer is always possible In any type of confocal microscope or focus variation microscope is not difficult to measure the radius of a sphere. Al least, it is possible to measure the spherical surface, export the x,y,z coordinates in a text file and, after that, process the file with a routine like that included in Appendix A (new in the second version of the manuscript). Usually, the software permits to perform roughness measurements. A small number of roughness parameters can only be computed, but usually, it is always possible to work with Ra.

It’s true that to apply corrections the software of the confocal microscope should permit to introduce voxel dimensions in any way. But this is usually possible. Generally the software permits to introduce a value for pixel size ( and a parameter to correct the voxel height or z-readings. The most usual problem is that software usually assumes pixel size is the same along the X and Y axes. So, the calibration correction cannot be applied. The same problem appears with perpendicularity error. But this problem has been considered in the manuscript and a possible alternative has been proposed.

It’s true that our software permits to correct the flatness deviation of the focal plane, and this is usually not possible in other software. But, when a significant flatness deviation detected in a confocal microscope some action should be taken:

Estimate the flatness deviation and apply the correction. This can be done automatically if the software permits the introduction of the flatness deviation. Alternatively, it can be done by hand: exporting the x,y,z coordinates to a text file and then applying the flatness correction. If there is no possibility of the introduction of the flatness correction, the user should ask the manufacturer of the microscope to adjust and physically correct the flatness deviation up to a level, where the flatness deviation is negligible. If the flatness deviation is significant and it can not be solved (no correction can be applied nor adjustment can be made), then the confocal microscope can not be used as a metrological instrument, at least for measurements where coordinate z is present. Therefore, it could be a good idea that confocal users press the manufacturers to introduce the flatness correction in their software.

We hope that the modifications we made in the abstract, the introduction and the conclusions make clearer that this is not a particular procedure for a particular type or model, but it is a general one. Even, we have included in Appendix A the source code of a routine to make a least-square fit of a sphere in order to make clearer that our objective is to make the application of our procedure as wider as possible.

It is true that figures 1 and 2 are not strictly necessary because the working principle of the confocal microscope is described in the text. But we considered that, sometimes, a bad image is better to communicate an idea that a good text. That is the reason because we have maintained figures 1 and 2.

Following your comments:

We have checked all the figures of the document and we think that, in the second version of the manuscript, all the figures are cited in the body of the text. We have to improve the quality of figure 6. Now the readability of the text is better. We have corrected the captions of figure 13 and table 3

Sincerely yours,

Alberto Mínguez

Jesús de Vicente

Round 2

Reviewer 1 Report

All the issues were included. No further comments